# 8-bit Optimizers via Block-wise Quantization

## Abstract

Stateful optimizers maintain gradient statistics over time, e.g., the exponentially smoothed sum (SGD with momentum) or squared sum (Adam) of past gradient values. This state can be used to accelerate optimization compared to plain stochastic gradient descent but uses memory that might otherwise be allocated to model parameters, thereby limiting the maximum size of models trained in practice. In this paper, we develop the first optimizers that use 8-bit statistics while maintaining the performance levels of using 32-bit optimizer states. To overcome the resulting computational, quantization, and stability challenges, we develop block-wise dynamic quantization. Block-wise quantization divides input tensors into smaller blocks that are independently quantized. Each block is processed in parallel across cores, yielding faster optimization and high precision quantization. To maintain stability and performance, we combine block-wise quantization with two additional changes: (1) dynamic quantization, a form of non-linear optimization that is precise for both large and small magnitude values, and (2) a stable embedding layer to reduce gradient variance that comes from the highly non-uniform distribution of input tokens in language models. As a result, our 8-bit optimizers maintain 32-bit performance with a small fraction of the memory footprint on a range of tasks, including 1.5B parameter language modeling, GLUE finetuning, ImageNet classification, WMT'14 machine translation, MoCo v2 contrastive ImageNet pretraining+finetuning, and RoBERTa pretraining, without changes to the original optimizer hyperparameters. We open-source our 8-bit optimizers as a drop-in replacement that only requires a two-line code change.

Increasing model size is an effective way to achieve better performance for given resources (Kaplan et al., 2020; Henighan et al., 2020; Raffel et al., 2019; Lewis et al., 2021). However, training such large models requires storing the model, gradient, and state of the optimizer (e.g., exponentially smoothed sum and squared sum of previous gradients for Adam), all in a fixed amount of available memory. Although significant research has focused on enabling larger model training by reducing or efficiently distributing the memory required for the model parameters (Shoeybi et al., 2019; Lepikhin et al., 2020; Fedus et al., 2021; Brown et al., 2020; Rajbhandari et al., 2020), reducing the memory footprint of optimizer gradient statistics is much less studied. This is a significant missed opportunity since these optimizer states use 33-75% of the total memory footprint during training. For example, the Adam optimizer states for the largest GPT-2 (Radford et al., 2019) and T5 (Raffel et al., 2019) models are 11 GB and 41 GB in size. In this paper, we develop a fast, high-precision non-linear quantization method – block-wise dynamic quantization – that enables stable 8-bit optimizers (e.g., Adam, AdamW, and Momentum) which maintain 32-bit performance at a fraction of the memory footprint and without any changes to the original hyperparameters.[1]

While most current work uses 32-bit optimizer states, recent high-profile efforts to use 16-bit optimizers report difficultly for large models with more than 1B parameters (Ramesh et al., 2021). Going from 16-bit optimizers to 8-bit optimizers reduces the range of possible values from $2^{16} = 65536$ values to just $2^8 = 256$. To our knowledge, this has not been attempted before.

Effectively using this very limited range is challenging for three reasons: quantization accuracy, computational efficiency, and large-scale stability. To maintain accuracy, it is critical to introduce some form of non-linear quantization to reduce errors for both common small magnitude values and rare large ones. However, to be practical, 8-bit optimizers need to be fast enough to not slow

---

[1]We study 8-bit optimization with current best practice model and gradient representations (typically 16-bit mixed precision), to isolate optimization challenges. Future work could explore further compressing all three.

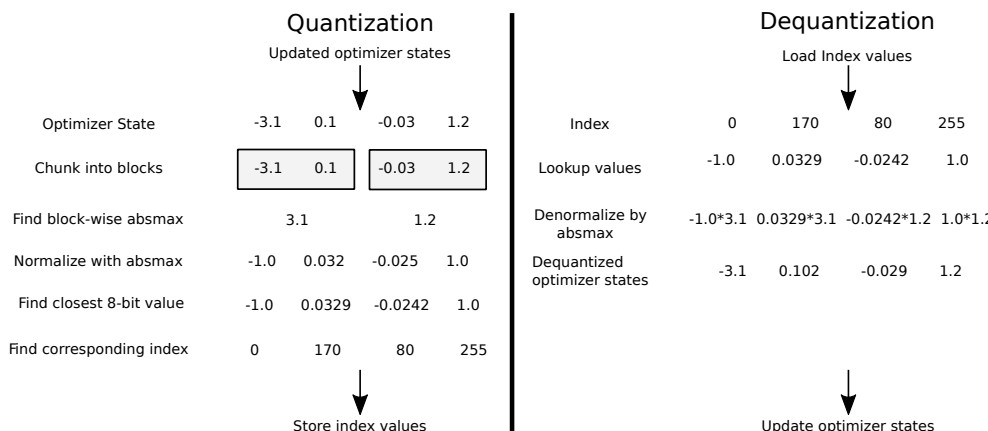

Figure 1: Schematic of 8-bit optimizers via block-wise dynamic quantization, see Section 2 for more details. After the optimizer update is performed in 32-bit, the state tensor is chunked into blocks, normalized by the absolute maximum value of each block. Then dynamic quantization is performed, and the index is stored. For dequantization, a lookup in the index is performed, with subsequent denormalization by multiplication with the block-wise absolute maximum value. Outliers are confined to a single block through block-wise quantization, and their effect on normalization is limited.

down training, which is especially difficult for non-linear methods that require more complex data structures to maintain the quantization buckets. Finally, to maintain stability with huge models beyond 1B parameters, a quantization method needs to not only have a good mean error but excellent worse case performance since a single large quantization error can cause the entire training run to diverge.

We introduce a new block-wise quantization approach that addresses all three of these challenges. Block-wise quantization splits input tensors into blocks and performs quantization on each block independently. This block-wise division reduces the effect of outliers on the quantization process since they are isolated to particular blocks, thereby improving stability and performance, especially for large-scale models. Block-wise processing also allows for high optimizer throughput since each normalization can be computed independently in each core. This contrasts with tensor-wide normalization, which requires slow cross-core synchronization that is highly dependent on task-core scheduling. We combine block-wise quantization with two novel methods for stable, high-performance 8-bit optimizers: dynamic quantization and a stable embedding layer. Dynamic quantization is an extension of dynamic tree quantization for unsigned input data. The stable embedding layer is a variation of a standard word embedding layer that supports more aggressive quantization by normalizing the highly non-uniform distribution of inputs to avoid extreme gradient variation.

Our 8-bit optimizers maintain 32-bit performance at a fraction of the original memory footprint. We show this for a broad range of tasks: 1.5B and 355M parameter language modeling, GLUE finetuning, ImageNet classification, WMT'14+WMT'16 machine translation, MoCo v2 contrastive image pretraining+finetuning, and RoBERTa pretraining. We also report additional ablations and sensitivity analysis showing that all components – block-wise quantization, dynamic quantization, and stable embedding layer – are crucial for these results and that 8-bit Adam can be used as a simple drop-in replacement for 32-bit Adam, with no hyperparameter changes. We open-source our custom CUDA kernels and provide a PyTorch implementation that enables 8-bit optimization by changing two lines of code.

# 1 BACKGROUND

## 1.1 STATEFUL OPTIMIZERS

An optimizer updates the parameters $\mathbf{w}$ of a neural network by using the gradient of the loss with respect to the weight $\mathbf{g}_t = \frac{\partial \mathbf{L}}{\partial \mathbf{w}}$ at update iteration $t$. Stateful optimizers compute statistics of the gradient with respect to each parameter over time for accelerated optimization. Two of the most commonly used stateful optimizers are Adam (Kingma and Ba, 2014), and SGD with momentum

(Qian, 1999) – or Momentum for short. Without damping and scaling constants, the update rules of these optimizers are given by:

$$\text{Momentum}(\mathbf{g}_t, \mathbf{w}_{t-1}, \mathbf{m}_{t-1}) = \begin{cases} \mathbf{m}_0 = \mathbf{g}_0 & \text{Initialization} \\ \mathbf{m}_t = \beta_1 \mathbf{m}_{t-1} + \mathbf{g}_t & \text{State 1 update} \\ \mathbf{w}_t = \mathbf{w}_{t-1} - \alpha \cdot \mathbf{m}_t & \text{Weight update} \end{cases} \tag{1}$$

$$\text{Adam}(\mathbf{g}_t, \mathbf{w}_{t-1}, \mathbf{m}_{t-1}, \mathbf{r}_{t-1}) = \begin{cases} \mathbf{r}_0 = \mathbf{m}_0 = \mathbf{0} & \text{Initialization} \\ \mathbf{m}_t = \beta_1 \mathbf{m}_{t-1} + (1 - \beta_1)\mathbf{g}_t & \text{State 1 update} \\ \mathbf{r}_t = \beta_2 \mathbf{r}_{t-1} + (1 - \beta_2)\mathbf{g}_t^2 & \text{State 2 update} \\ \mathbf{w}_t = \mathbf{w}_{t-1} - \alpha \cdot \frac{\mathbf{m}_t}{\sqrt{\mathbf{r}_t} + \epsilon} & \text{Weight update,} \end{cases} \tag{2}$$

where $\beta_1$ and $\beta_2$ are smoothing constants, $\epsilon$ is a small constant, and $\alpha$ is the learning rate.

For 32-bit states, Momentum and Adam consume 4 and 8 bytes per parameter. That is 4 GB and 8 GB for a 1B parameter model. Our 8-bit non-linear quantization reduces these costs to 1 GB and 2 GB.

## 1.2 NON-LINEAR QUANTIZATION

Quantization compresses numeric representations to save space at the cost of precision. Quantization is the mapping of a $k$-bit integer to a real element in $D$, that is, $\mathbf{Q}^{\text{map}} : [0, 2^k - 1] \mapsto D$. For example, the IEEE 32-bit floating point data type maps the indices $0...2^{32} - 1$ to the domain [-3.4e38, +3.4e38]. We use the following notation: $\mathbf{Q}^{\text{map}}(i) = \mathbf{Q}_i^{\text{map}} = q_i$, for example $\mathbf{Q}^{\text{map}}(2^{31} + 131072) = 2.03125$, for the IEEE 32-bit floating point data type.

To perform general quantization from one data type into another we require three steps. (1) Compute a normalization constant $N$ that transforms the input tensor $\mathbf{T}$ into the range of the domain $D$ of the target quantization data type $\mathbf{Q}^{\text{map}}$, (2) for each element of $\mathbf{T}/N$ find the closest corresponding value $q_i$ in the domain $D$, (3) store the index $i$ corresponding to $q_i$ in the quantized output tensor $\mathbf{T}^Q$. To receive the dequantized tensor $\mathbf{T}^D$ we look up the index and denormalize: $\mathbf{T}_i^D = \mathbf{Q}^{\text{map}}(\mathbf{T}_i^Q) \cdot N$.

To perform this procedure for dynamic quantization we first normalize into the range [-1, 1] through division by the absolute maximum value: $N = \max(|\mathbf{T}|)$.

Then we find the closest values via a binary search:

$$\mathbf{T}_i^Q = \underset{j=0}{\overset{2^n}{\arg\min}} |\mathbf{Q}_j^{\text{map}} - \frac{\mathbf{T}_i}{N}| \tag{3}$$

## 1.3 DYNAMIC TREE QUANTIZATION

Dynamic Tree quantization (Dettmers, 2016) is a method that yields low quantization error for both small and large magnitude values. Unlike data types with fixed exponent and fraction, dynamic tree quantization uses a datatype with a dynamic exponent and fraction that can change with each number. It is made up of four parts, as seen in Figure 2: (1) The first bit of the data type is reserved for a sign. (2) The number of subsequent zero bits indicates the magnitude of the exponent. (3) The first bit that is set to one indicates that all following values are reserved for (4) linear quantization. By moving the indicator bit, num-

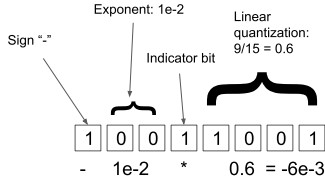

Figure 2: Dynamic tree quantization.

bers can have a large exponent $10^{-7}$ or precision as high as $1/63$. Compared to linear quantization, dynamic tree quantization has better absolute and relative quantization errors for non-uniform distributions. Dynamic tree quantization is strictly defined to quantize numbers in the range [-1.0, 1.0], which is ensured by performing tensor-level absolute max normalization.

## 2    8-BIT OPTIMIZERS

Our 8-bit optimizers have three components: (1) block-wise quantization that isolates outliers and distributes the error more equally over all bits; (2) dynamic quantization, which quantizes both small and large values with high precision; and (3) a stable embedding layer to improve stability during optimization for models with word embeddings.

With these components, performing an optimizer update with 8-bit states is straightforward. We dequantize the 8-bit optimizer states to 32-bit, perform the update, and then quantize the states back to 8-bit for storage. We do this 8-bit to 32-bit conversion element-by-element in registers, which means no slow copies to GPU memory or additional temporary memory are needed to perform quantization and dequantization. For GPUs, this makes 8-bit optimizers faster than regular 32-bit optimizers, as we show in Section 3.

### 2.1    BLOCK-WISE QUANTIZATION

Our block-wise quantization reduces the cost of computing normalization and improves quantization precision by isolating outliers. In order to dynamically quantize a tensor, as defined in Section 1.2, we need to normalize the tensor into the range [-1, 1]. Such normalization requires a reduction over the entire tensor, which entails multiple synchronizations across GPU cores. Block-wise dynamic quantization reduces this cost by chunking an input tensor into small blocks of size $B = 2048$ and performing normalization independently in each core across this block.

More formally, using the notation introduced in Section 1.2, in block-wise quantization, we treat $\mathbf{T}$ as a one-dimensional sequence of elements that we chunk in blocks of size $B$. This means for an input tensor $\mathbf{T}$ with $n$ elements we have $n/B$ blocks. We proceed to compute a normalization constant for each block: $N_b = \max(|\mathbf{T}_b|)$, where $b$ is the index of the block $0..n/B$. With this block-wise normalization constant, each block can be quantized independently:

$$\mathbf{T}_{bi}^Q = \underset{j=0}{\arg\min}^{2^n} \left| \mathbf{Q}_j^{\text{map}} - \frac{\mathbf{T}_{bi}}{N_b} \right| \Big|_{0<i<B} \tag{4}$$

This approach has several advantages, both for stability and efficiency. First, each block normalization can be computed independently. Thus no synchronization between cores is required, and throughput is enhanced.

Secondly, it is also much more robust to outliers in the input tensor. For example, to contrast block-wise and regular quantization, if we create an input tensor with one million elements sampled from the standard normal distribution, we expect less than 1% of elements of the tensor will be in the range $[3, +\infty)$. However, since we normalize the input tensor into the range [-1,1] this means the maximum values of the distribution determine the range of quantization buckets. This means if the input tensor contains an outlier with magnitude 5, the quantization buckets reserved for numbers between 3 and 5 will mostly go unused since less than 1% of numbers are in this range. With block-wise quantization, the effect of outliers is limited to a single block. As such, most bits are used effectively in other blocks.

Furthermore, because outliers represent the absolute maximum value in the input tensor, block-wise quantization approximates outlier values without any error. This guarantees that the largest optimizer states, arguably the most important, will always be quantized with full precision. This property makes block-wise dynamic quantization both robust and precise and is essential for good training performance in practice.

### 2.2    DYNAMIC QUANTIZATION

In this work, we extend dynamic tree quantization (Section 1.3) for non-signed input tensors by re-purposing the sign bit. Since the second Adam state is strictly positive, the sign bit is not needed. Instead of just removing the sign bit, we opt to extend dynamic tree quantization with a *fixed* bit for the fraction. This extension is motivated by the observation that the second Adam state varies around 3-5 orders of magnitude during the training of a language model. In comparison, dynamic tree quantization already has a range of 7 orders of magnitude. We refer to this quantization as

*dynamic quantization* to distinguish it from dynamic tree quantization in our experiments. A study of additional quantization data types and their performance is detailed in Appendix E.

## 2.3 STABLE EMBEDDING LAYER

Our stable embedding layer is a standard word embedding layer variation (Devlin et al., 2019) designed to ensure stable training for NLP tasks. This embedding layer supports more aggressive quantization by normalizing the highly non-uniform distribution of inputs to avoid extreme gradient variation. See Appendix B for a discussion of why commonly adopted embedding layers (Ott et al., 2019) are so unstable.

We initialize the Stable Embedding layer with Xavier uniform initialization (Glorot and Bengio, 2010) and apply layer normalization (Ba et al., 2016) before adding position embeddings. This method maintains a variance of roughly one both at initialization and during training. Additionally, the uniform distribution initialization has less extreme values than a normal distribution, reducing maximum gradient size. Like Ramesh et al. (2021), we find that the stability of training improves significantly if we use 32-bit optimizer states for the embedding layers. This is the only layer that uses 32-bit optimizer states. We still use the standard precision for weights and gradients for the embedding layers – usually 16-bit. We show in our Ablation Analysis in Section 4 that this change is a necessary detail.

## 3 8-BIT VS 32-BIT OPTIMIZER PERFORMANCE FOR COMMON BENCHMARKS

**Experimental Setup** We compare the performance of 8-bit optimizers to their 32-bit counterparts on a range of challenging public benchmarks. These benchmarks either use Adam (Kingma and Ba, 2014), AdamW (Loshchilov and Hutter, 2018), or Momentum (Qian, 1999).

We do not change any hyperparameters or precision of weights, gradients, and activations/input gradients for each experimental setting compared to the public baseline– the only change is to replace 32-bit optimizers with 8-bit optimizers. This means that for most experiments, we train in 16-bit mixed-precision (Micikevicius et al., 2017). We also compare with Adafactor (Shazeer and Stern, 2018), with the time-independent formulation for $\beta_2$ (Shazeer and Stern, 2018) – which is the same formulation used in Adam. We also do not change any hyperparameters for Adafactor.

We report on benchmarks in neural machine translation (Ott et al., 2018)[2] trained on WMT'16 (Sennrich et al., 2016) and evaluated on en-de WMT'14 (Macháček and Bojar, 2014), large-scale language modeling (Lewis et al., 2021; Brown et al., 2020) and RoBERTa pretraining (Liu et al., 2019) on English CC-100 + RoBERTa corpus (Nagel, 2016; Gokaslan and Cohen, 2019; Zhu et al., 2015; Wenzek et al., 2020), finetuning the pretrained masked language model RoBERTa (Liu et al., 2019)[3] on GLUE (Wang et al., 2018a), ResNet-50 v1.5 image classification (He et al., 2016)[4] on ImageNet-1k (Deng et al., 2009), and Moco v2 contrastive image pretraining and linear finetuning (Chen et al., 2020b)[5] on ImageNet-1k (Deng et al., 2009).

We use the stable embedding layer for all NLP tasks except for finetuning on GLUE. Beyond this, we follow the exact experimental setup outlined in the referenced papers and codebases. We consistently report replication results for each benchmark with public codebases and report median accuracy, perplexity, or BLEU over ten random seeds for GLUE, three random seeds for others tasks, and a single random seed for large scale language modeling. While it is standard to report means and standard errors on some tasks, others use median performance. We opted to report medians for all tasks for consistency.

**Results** In Table 1, we see that 8-bit optimizers match replicated 32-bit performance for all tasks. While Adafactor is competitive with 8-bit Adam, 8-bit Adam uses less memory and provides faster optimization. Our 8-bit optimizers save up to 8.5 GB of GPU memory for our largest 1.5B parameter language model and 2.0 GB for RoBERTa. Thus, 8-bit optimizers maintain performance

---

[2] `https://github.com/pytorch/fairseq/tiny/master/examples/scaling_nmt/README.md`

[3] `https://github.com/pytorch/fairseq/blob/master/examples/roberta/README.glue.md`

[4] `https://github.com/NVIDIA/DeepLearningExamples/tree/master/PyTorch/Classification/ConvNets/`

[5] `https://github.com/facebookresearch/moco`

Table 1: Median performance on diverse NLP and computer vision tasks: GLUE, object classification with (Moco v2) and without pretraining (CLS), machine translation (MT), and large-scale language modeling (LM). While 32-bit Adafactor is competitive with 8-bit Adam, it uses almost twice as much memory and trains slower. 8-bit Optimizers match or exceed replicated 32-bit performance on all tasks. We observe no instabilities for 8-bit optimizers. Time is total GPU time on V100 GPUs, except for RoBERTa and GPT3 pretraining, which were done on A100 GPUs.

| Optimizer | Task | Data | Model | Metric† | Time | Mem saved |
|---|---|---|---|---|---|---|
| 32-bit AdamW | GLUE | Multiple | RoBERTa-Large | 88.9 | – | Reference |
| 32-bit AdamW | GLUE | Multiple | RoBERTa-Large | 88.6 | 17h | 0.0 GB |
| 32-bit Adafactor | GLUE | Multiple | RoBERTa-Large | **88.7** | 24h | 1.3 GB |
| 8-bit AdamW | GLUE | Multiple | RoBERTa-Large | **88.7** | **15h** | **2.0 GB** |
| 32-bit Momentum | CLS | ImageNet-1k | ResNet-50 | 77.1 | – | Reference |
| 32-bit Momentum | CLS | ImageNet-1k | ResNet-50 | 77.1 | 118h | 0.0 GB |
| 8-bit Momentum | CLS | ImageNet-1k | ResNet-50 | **77.2** | **116 h** | **0.1 GB** |
| 32-bit Adam | MT | WMT'14+16 | Transformer | 29.3 | – | Reference |
| 32-bit Adam | MT | WMT'14+16 | Transformer | 29.0 | 126h | 0.0 GB |
| 32-bit Adafactor | MT | WMT'14+16 | Transformer | 29.0 | 127h | 0.3 GB |
| 8-bit Adam | MT | WMT'14+16 | Transformer | **29.1** | **115h** | **1.1 GB** |
| 32-bit Momentum | MoCo v2 | ImageNet-1k | ResNet-50 | 67.5 | – | Reference |
| 32-bit Momentum | MoCo v2 | ImageNet-1k | ResNet-50 | 67.3 | 30 days | 0.0 GB |
| 8-bit Momentum | MoCo v2 | ImageNet-1k | ResNet-50 | **67.4** | **28 days** | **0.1 GB** |
| 32-bit Adam | LM | Multiple | Transformer-1.5B | 9.0 | 308 days | 0.0 GB |
| 32-bit Adafactor | LM | Multiple | Transformer-1.5B | **8.9** | 316 days | 5.6 GB |
| 8-bit Adam | LM | Multiple | Transformer-1.5B | 9.0 | **297 days** | **8.5 GB** |
| 32-bit Adam | LM | Multiple | GPT3-Medium | 10.62 | 795 days | 0.0 GB |
| 32-bit Adafactor | LM | Multiple | GPT3-Medium | 10.68 | 816 days | 1.5 GB |
| 8-bit Adam | LM | Multiple | GPT3-Medium | **10.62** | **761 days** | **1.7 GB** |
| 32-bit Adam | Masked-LM | Multiple | RoBERTa-Base | 3.49 | 101 days | 0.0 GB |
| 32-bit Adafactor | Masked-LM | Multiple | RoBERTa-Base | 3.59 | 112 days | 0.7 GB |
| 8-bit Adam | Masked-LM | Multiple | RoBERTa-Base | **3.48** | **94 days** | **1.1 GB** |

†**Metric**: GLUE=Mean Accuracy/Correlation. CLS/MoCo = Accuracy. MT=BLEU. LM=Perplexity.

and improve accessibility to the finetuning of large models for those that cannot afford GPUs with large memory buffers. We show models that are now accessible with smaller GPUs in Table 2. A breakdown of individual dataset results on GLUE can be found in Appendix A).

The broad range of tasks and competitive results demonstrate that 8-bit optimizers are a robust and effective replacement for 32-bit optimizers, do not require any additional changes in hyperparameters, and save a significant amount of memory while speeding up training slightly.

Table 2: With 8-bit optimizers, larger models can be finetuned with the same GPU memory compared to standard 32-bit optimizer training. We use a batch size of one for this comparison.

| | Largest finetunable Model (parameters) | |
|---|---|---|
| GPU size in GB | 32-bit Adam | 8-bit Adam |
| 6 | RoBERTa-base (110M) | RoBERTa-large (355M) |
| 11 | MT5-small (300M) | MT5-base (580M) |
| 24 | MT5-base (580M) | MT5-large (1.2B) |
| 24 | GPT-2-medium (762M) | GPT-2-large (1.5B) |

## 4 ANALYSIS

We analyze our method in two ways. First, we ablate all 8-bit optimizer components and show that they are necessary for good performance. Second, we look at the sensitivity to hyperparameters

compared to 32-bit Adam and show that 8-bit Adam with block-wise dynamic quantization is a reliable replacement that does not require further hyperparameter tuning.

**Experimental Setup** We perform our analysis on a strong 32-bit Adam baseline for language modeling with transformers (Vaswani et al., 2017). We subsample from the RoBERTa corpus (Liu et al., 2019) which consists of the English sub-datasets: Books (Zhu et al., 2015), Stories (Trinh and Le, 2018), OpenWebText-1 (Gokaslan and Cohen, 2019), Wikipedia, and CC-News (Nagel, 2016). We use a 50k token BPE encoded vocabulary (Sennrich et al., 2015). We find the best 2-GPU-day transformer baseline for 32-bit Adam with multiple hyperparameter searches that take in a total of 440 GPU days. Key hyperparameters include 10 layers with a model dimension of 1024, a fully connected hidden dimension of 8192, 16 heads, and input sub-sequences with a length of 512 tokens each. The final model has 209m parameters.

Table 3: Ablation analysis of 8-bit Adam for small (2 GPU days) and large-scale ($\approx$1 GPU year) transformer language models on the RoBERTa corpus. The runs without dynamic quantization use linear quantization. The percentage of unstable runs indicates either divergence or crashed training due to exploding gradients. We report median perplexity for successful runs. We can see that dynamic quantization is critical for general stability and block-wise quantization is critical for large-scale stability. The stable embedding layer is useful for both 8-bit and 32-bit Adam and enhances stability to some degree.

| Parameters | Optimizer | Dynamic | Block-wise | Stable Emb | Unstable (%) | Perplexity |
|---|---|---|---|---|---|---|
| | 32-bit Adam | | | | 0 | 16.7 |
| | 32-bit Adam | | | ✓ | 0 | **16.3** |
| | 8-bit Adam | | | | 90 | 253.0 |
| | 8-bit Adam | | | ✓ | 50 | **194.4** |
| 209M | 8-bit Adam | ✓ | | | 10 | 18.6 |
| | 8-bit Adam | ✓ | | ✓ | 0 | **17.7** |
| | 8-bit Adam | ✓ | ✓ | | 0 | 16.8 |
| | 8-bit Adam | ✓ | ✓ | ✓ | 0 | **16.4** |
| 1.3B | 32-bit Adam | | | | 0 | 10.4 |
| 1.3B | 8-bit Adam | ✓ | | | 100 | N/A |
| 1.3B | 8-bit Adam | ✓ | | ✓ | 80 | 10.9 |
| 1.5B | 32-bit Adam | | | | 0 | 9.0 |
| 1.5B | 8-bit Adam | ✓ | ✓ | ✓ | 0 | 9.0 |

**Ablation Analysis** For the ablation analysis, we compare small and large-scale language modeling perplexity and training stability against a 32-bit Adam baseline. We ablate components individually and include combinations of methods that highlight their interactions. The baseline method uses linear quantization, and we add dynamic quantization, block-wise quantization, and the stable embedding layer to demonstrate their effect. To test optimization stability for small-scale language modeling, we run each setting with different hyperparameters and report median performance across all successful runs. A successful run is a run that does not crash due to exploding gradients or diverges in the loss. We use the hyperparameters $\epsilon$ {1e-8, 1e-7, 1e-6}, $\beta_1$ {0.90, 0.87, 0.93}, $\beta_2$ {0.999, 0.99, 0.98} and small changes in learning rates. We also include some partial ablations for large-scale models beyond 1B parameters. In the large-scale setting, we run several seeds with the same hyperparameters. We use a single seed for 32-bit Adam, five seeds for 8-bit Adam at 1.3B parameters, and a single seed for 8-bit Adam at 1.5B parameters.[6] Results are shown in Table 3.

The Ablations show that dynamic quantization, block-wise quantization, and the stable embedding layer are critical for either performance or stability. In addition, block-wise quantization is critical for large-scale language model stability.

**Sensitivity Analysis** We compare the perplexity of 32-bit Adam vs 8-bit Adam + Stable Embedding as we change the optimizer hyperparameters: learning rate, betas, and $\epsilon$. We change each hyperparameter individually from the baseline hyperparameters $\beta_1$=0.9, $\beta_2$=0.995, $\epsilon$=1e-7, and lr=0.0163

---

[6]We chose not to do the full ablations with such large models because each training run takes one GPU year.

and run two random seeds for both 8-bit and 32-bit Adam for each setting. If 8-bit Adam is perfectly insensitive to hyperparameters compared to 32-bit Adam, we would expect the same constant offset in performance for any hyperparameter combination. The results can be seen in Figure 3. The results show a relatively steady gap between 8-bit and 32-bit Adam, suggesting that 8-bit Adam does not require any further hyperparameter tuning compared to 32-bit Adam.

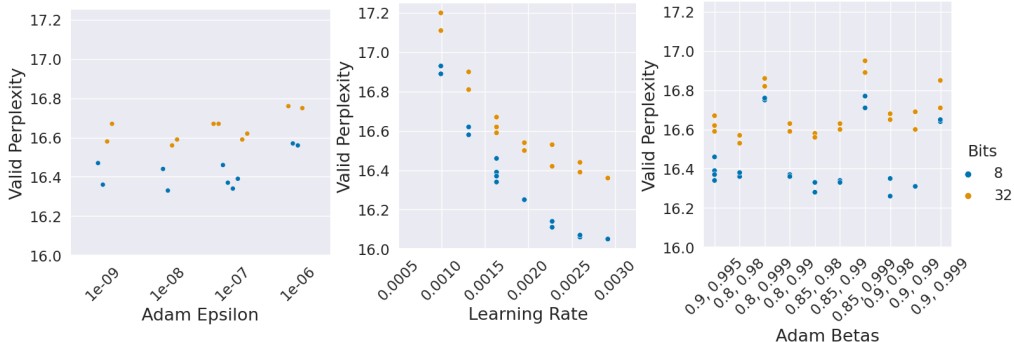

Figure 3: Sensitivity analysis of 8-bit vs 32-bit Adam hyperparameters. We can see that there is little variance between 8 and 32-bit performance, which suggests that 8-bit Adam can be used as a drop-in replacement for 32-bit Adam without any further hyperparameter tuning.

## 5 RELATED WORK

**Compressing & Distributing Optimizer States** While 16-bit Adam has been used in several publications, the stability of 16-bit Adam was first explicitly studied for a text-to-image generation model DALL-E (Ramesh et al., 2021). They show that a stable embedding layer, tensor-wise scaling constants for both Adam states, and multiple loss scaling blocks are critical to achieving stability during training. Our work reduces the memory footprint of Adam further, from 16 to 8-bit. In addition, we achieve stability by developing new training procedures and non-linear quantization, both of which complement previous developments.

Adafactor (Shazeer and Stern, 2018) uses a different strategy to save memory. All optimizer states are still 32-bit, but the second Adam state is factorized by a row-column outer product resulting in a comparable memory footprint to 16-bit Adam. Alternatively, Adafactor can also be used without using the first moment ($\beta_1 = 0.0$) (Lepikhin et al., 2020). This version is as memory efficient as 8-bit Adam, but unlike 8-bit Adam, hyperparameters for this Adafactor variant need to be re-tuned to achieve good performance. We compare 8-bit Adam with Adafactor $\beta_1 > 0.0$ in our experiments.

AdaGrad (Duchi et al., 2011) adapts the gradient with aggregate training statistics over the entire training run. AdaGrad that uses only the main diagonal as optimizer state and extensions of AdaGrad such as SM3 (Anil et al., 2019) and extreme tensoring (Chen et al., 2020a) can be more efficient than 8-bit Adam. We include some initial comparison with AdaGrad in Appendix G.

Optimizer sharding (Rajbhandari et al., 2020) splits optimizer states across multiple accelerators such as GPUs/TPUs. While very effective, it can only be used if multiple accelerators are available and data parallelism is used. Optimizer sharding can also have significant communication overhead (Rajbhandari et al., 2021). Our 8-bit optimizers work with all kinds of parallelism. They can also complement optimizer sharding, as they reduce communication overhead by 75%.

**General Memory Reduction Techniques** Other complementary methods for efficient training can be either distributed or local. Distributed approaches spread out the memory of a model across several accelerators such as GPUs/TPUs. Such approaches are model parallelism (Krizhevsky et al., 2009), pipeline parallelism (Krizhevsky et al., 2009; Huang et al., 2018; Harlap et al., 2018), and operator parallelism (Lepikhin et al., 2020). These approaches are useful if one has multiple accelerators available. Our 8-bit optimizers are useful for both single and multiple devices.

Local approaches work for a single accelerator. They include gradient checkpointing (Chen et al., 2016), reversible residual connections (Gomez et al., 2017), and offloading (Pudipeddi et al., 2020;

Rajbhandari et al., 2021). All these methods save memory at the cost of increased computational or communication costs. Our 8-bit optimizers reduce the memory footprint of the model while maintaining 32-bit training speed.

**Quantization Methods and Data Types** While our work is the first to apply 8-bit quantization to optimizer statistics, quantization for neural network model compression, training, and inference are well-studied problems. One of the most common formats of 8-bit quantization is to use data types composed of static sign, exponent, and fraction bits. The most common combination is 5 bits for the exponent and 2 bits for the fraction (Wang et al., 2018b; Sun et al., 2019; Cambier et al., 2020; Mellempudi et al., 2019) with either no normalization or min-max normalization. These data types offer high precision for small magnitude values but have large errors for large magnitude values since only 2 bits are assigned to the fraction. Other methods improve quantization through soft constraints (Li et al., 2021) or more general uniform affine quantizations (Pappalardo, 2021).

Data types lower than 8-bit are usually used to prepare a model for deployment, and the main focus is on improving network inference speed and memory footprint rather than maintaining accuracy. There are methods that use 1-bit (Courbariaux and Bengio, 2016; Rastegari et al., 2016; Courbariaux et al., 2015), 2-bit/3 values (Zhu et al., 2017; Choi et al., 2019), 4-bits (Li et al., 2019), more bits (Courbariaux et al., 2014), or a variable amount of bits (Gong et al., 2019). See also Qin et al. (2020) for a survey on binary neural networks. While these low-bit quantization techniques allow for efficient storage, they likely lead to instability when used for optimizer states.

The work most similar to our block-wise quantization is work on Hybrid Block Floating Point (HBFP) (Drumond et al., 2018) which uses a 24-bit fraction data type with a separate exponent for each tile in matrix multiplication to perform 24-bit matrix multiplication. However, unlike HBFP, block-wise dynamic quantization has the advantage of having both block-wise normalization *and* a dynamic exponent for each number. This allows for a much broader range of important values since optimizer state values vary by about 5 orders of magnitude. Furthermore, unlike HBFP, block-wise quantization approximates the maximum magnitude values within each block without any quantization error, which is critical for optimization stability, particularly for large networks.

## 6 DISCUSSION & LIMITATIONS

Here we have shown that high precision quantization can yield 8-bit optimizers that maintain 32-bit optimizer performance without requiring any change in hyperparameters. One of the main limitations of our work is that 8-bit optimizers for natural language tasks require a stable embedding layer to be trained to 32-bit performance. On the other hand, we show that 32-bit optimizers also benefit from a stable embedding layer. As such, the stable embedding layer could be seen as a general replacement for other embedding layers.

We show that 8-bit optimizers reduce the memory footprint and accelerate optimization on a wide range of tasks. However, since 8-bit optimizers reduce only the memory footprint proportional to the number of parameters, models that use large amounts of activation memory and little memory for parameters, such as convolutional networks, have few benefits from using 8-bit optimizers. Thus, 8-bit optimizers are most beneficial for training or finetuning models with many parameters on highly memory-constrained GPUs.

Furthermore, there remain sources of instability that, to our knowledge, are not well understood. For example, we observed that models with over 1B parameters often have hard systemic divergence, where many parameters simultaneously cause exploding gradients. In other cases, a single parameter among those 1B parameters assumed a value too large, caused an exploding gradient, and led to a cascade of instability. It might be that this rare, soft cascading instability is related to the phenomena where instability disappears after reloading a model checkpoint and rolling a new random seed – a method standard for training huge models. This cascading instability might also be related to the observation that the larger a model is, the more unstable it becomes. For our 8-bit optimizers, we primarily needed the stable embedding layer to avoid cascading instability. Thus the stable embedding layer could potentially be viewed as decreasing the probability of extreme outlier gradients. If such phenomena were better understood, it could lead to better 8-bit optimizers and more stable training in general.

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

# A  GLUE SCORE BREAKDOWN

Table 4 contains the breakdown of individual scores on the GLUE datasets.

Table 4: Breakdown of GLUE scores. Each column is the median of 10 random seeds. The mean is the mean over medians.

| Model | MNLI | QNLI | QQP | RTE | SST-2 | MRPC | CoLA | STS-B | Mean |
|---|---|---|---|---|---|---|---|---|---|
| 32-bit Adam | 90.40 | 94.85 | 92.2 | 84.5 | 96.40 | 90.1 | 67.41 | 93.03 | 88.61 |
| 32-bit Adafactor | 90.35 | 94.70 | 92.2 | 85.4 | 96.45 | 90.0 | 67.63 | 92.91 | 88.71 |
| 8-bit Adam | 90.30 | 94.70 | 92.2 | 85.9 | 96.40 | 90.3 | 67.20 | 92.87 | 88.73 |

# B  STABILITY OF EMBEDDING LAYERS

Highly variable gradients can lead to unpredictable optimization behavior and instability that manifests as divergence or exploding gradients. Low precision optimziers can amplify variance of gradient updates due to the noise introduced during quantization. While our 8-bit optimizers appear to be stable for convolutional networks, similar to Ramesh et al. (2021), we find that word embedding layers are a major source of instability.

The main instability from the word embedding layer comes from the fact that it is a sparse layer with non-uniform distribution of inputs which can produce maximum gradient magnitudes 100x larger than other layers. For dense layers, if given $n$ samples arranged into $k$ mini-batches the sum of gradients of all mini-batches is always the same independent of how the $n$ samples are arranged into $k$ mini-batches. For embedding gradients, this depends on the arrangement of samples into mini-batches. This is because most deep learning frameworks normalize the gradient by the number of total tokens in the mini-batch, rather than the frequency of each individual token. This approximation allows stable learning with a single learning rate rather than variable learning rates that depend on token frequency in each individual mini-batch. However a side-effect of this method is that the magnitude of gradients for a particular token can vary widely with batch sizes and between different mini-batches.

There are multiple recipes for initialization word embedding layers. One of the most common recipes used in all models trained with fairseq (Ott et al., 2019) such as RoBERTa (Liu et al., 2019), BART (Lewis et al., 2020), large NMT models (Ott et al., 2018), and sparse expert models (Lewis et al., 2021), is the following: Initialize the word embedding layer with $N(0, 1/\sqrt{k})$ where $k$ is the embedding size of the embedding layer and to scale the outputs by $\sqrt{k}$. This scheme has a variance of one at the start of training for the output distribution to ensure good gradient flow.

We find this approach to induce some instability for 8-bit optimizers. We develop the stable embedding layer to solve this instability problem.

While the full recipe for our stable embedding layer is new, components of it has been used before. The layer norm after the embedding has been used before in work such as Devlin et al. (2019) and Radford et al. (2019) and enhanced precision for this particular layer was used in Ramesh et al. (2021). As pointed out above, these elements are not standard and the stable embedding layer combines three aspects that are all important: (1) enhanced precision, (2) layer norm, and (3) Xavier initialization.

# C  QUANTIZATION ERROR ANALYSIS

To gain more insights into why block-wise dynamic quantization works so well and how it could be improved, we performed a quantization error analysis of Adam quantization errors during language model training. Adam quantization errors are the deviations between the quantized 8-bit Adam update and the 32-bit Adam updates: $|\mathbf{u_8} - \mathbf{u_{16}}|$, where $\mathbf{u_k} = \mathbf{s_1^k}/\mathbf{s_2^k}$ for $k$ bits. See Background Section 1.1 for details on Adam.

A good 8-bit quantization has the property that, for a given input distribution, the inputs are only rarely quantized into intervals with high quantization error and most often quantized into intervals with low error.

In 8-bit, there are $255 \times 256$ possible 8-bit Adam updates, 256 possible values for the first and 256 for the second Adam state. We look at the average quantization error of each of these possible updates to see *where* the largest errors are and we plot histograms to see *how often* do these values with high error occur. Taken together, these two perspectives give a detailed view of the magnitude of deviations and how often large deviations occur.

We study these questions by looking at how often each of the 256 values for both Adam states are used during language model training. We also analyze the average error for each of the inputs quantized to each of the 256 values. With this analysis it is easy to find regions of high use and high error, and visualize their overlap. An overlap of these regions is associated with large frequent errors that cause unstable training. The quantization error analysis is shown in Figure 4.

The plots show two things: (1) The region of high usage (histogram) shows how often each combination of $256 \times 256$ bit values is used for the first Adam state $\mathbf{s_1}$ (exponentially smoothed running sum) and the second Adam state $\mathbf{s_2}$ (exponentially smoothed running squared sum). (2) The error plots show for $k$-bit Adam updates $\mathbf{u_k} = \mathbf{s_1}/(\sqrt{\mathbf{s_2}} + \epsilon)$ the mean absolute Adam error $|u_{32} - u_8|$ and the relative Adam error $|u_{32} - u_8|/|u_{32}|$ *averaged over each bit combination*. In conjunction these plots show which bits have the highest error per use and how often each bit is used. The x-axis/y-axis represents the quantization type range which means the largest positive/negative Adam states per block/tensor take the values 1.0/-1.0.

We can see that block-wise dynamic quantization has the smallest overlap between regions of high use and high error. While the absolute Adam quantization error of block-wise dynamic quantization is 0.0061, which is not much lower than that of dynamic quantization with 0.0067, the plots can also be interpreted as block-wise dynamic having rarer large errors that likely contribute to improved stability during optimization.

## D  Fine-grained Optimizer Runtime Performance

Table 5 shows optimizer performance that is benchmarked in isolation without any training. We use a large sample of a normal distribution and benchmark the average time to perform 100 optimizer updates per billion parameters in milliseconds.

Table 5: Runtime performance of 8-bit optimizers vs commonly used 32-bit optimizers in milliseconds per update per 1B parameters for 32-bit gradients. This comparision was run on a V100 GPU.

| Optimizer | Milliseconds per update per 1B param | | |
|---|---|---|---|
| | 32-bit PyTorch | 32-bit Apex | 8-bit (Ours) |
| Adam | 145 | 63 | **47** |
| Momentum | 58 | 46 | **34** |
| LAMB | – | 91 | **65** |
| LARS | – | 119 | **43** |

## E  Additional Quantization Data Types

This section describes additional quantization data types that we tried but which we found to perform poorly in quantization performance or stability. While quantile quantization has an average quantization twice as low as dynamic quantization for any normal distribution it has sporadic large errors that lead to large Adam errors and poor model performance (see Figure 5) and even with state-of-the-art quantile estimation algorithms (see Section F) quantile quantization is too slow to be practical. An overview of quantization performance of this additional quantization data types compared to dynamic quantization (without block-wise quantization) can be found in Table 6.

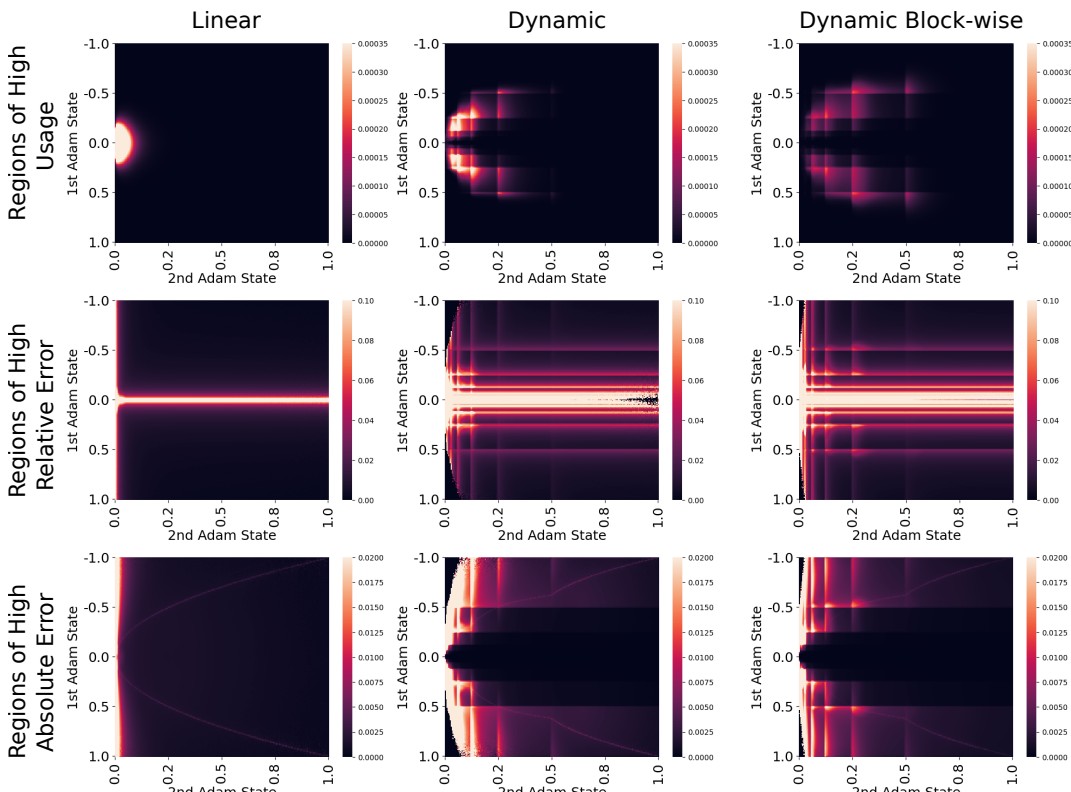

Figure 4: Good quantization methods do not have overlaps between regions of high use and high error. The plot shows that for linear quantization regions of high usage and high error overlap. For dynamic quantization regions with high relative error are used infrequently while only small regions have high usage and high absolute error. Block-wise dynamic quantization spreads out the usage over a large space and has the lowest overlap between regions of high use and errors. This means that not only is the overall error of block-wise dynamic quantization lower, but also that large errors for individual parameter updates are rarer compared to other methods, thus improving stability. See the main text for more details.

Table 6: Mean relative Adam and absolute quantization error for the first Adam state for different quantization methods. Results show mean±standard error. We can see that Dynamic Quantization has best relative error and that both Dynamic methods have the best absolute error.

| Method | Relative Adam Error | Absolute Quantization Error |
|---|---|---|
| Linear | 201% ±17% | 41.2e-10±3.1e-10 |
| Quantile | 11.9% ± 0.3% | 8.8e-10±0.9e-10 |
| Inverse Dynamic | 6.5%± 0.1% | **4.6e-10±0.4e-10** |
| Dynamic | **4.8%± 0.4%** | **3.5e-10±1.1e-10** |

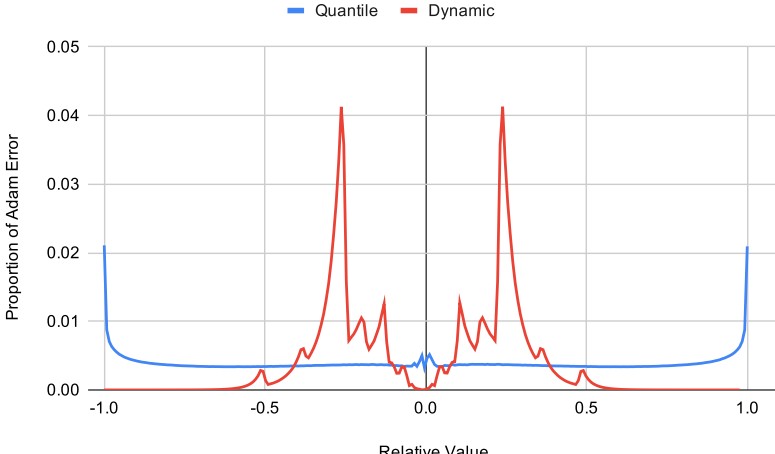

Figure 5: Distribution of Adam error among each of the 256 8-bit values of the first Adam state. We normalize the values into the range [-1,1]. With this, -1 indicates the largest negative value, 0 the value that is closest to 0, and so forth. See Figure 6 for a visualization of this normalization. Quantile quantization has large errors for large values, while dynamic quantization has small errors for both small and large values while the bulk of the errors is concentrated in intermediate values.

### E.1 INVERSE DYNAMIC QUANTIZATION

Inverse Dynamic Quantization is motivated by the hypothesis that large Adam updates are more important than small updates. Since Adam is composed of a ratio of optimizer states $\mathbf{m}_t/(\sqrt{\mathbf{r}_t}+\epsilon)$, we expect that small values in the second state $\mathbf{r}_t$ to produce large Adam updates. To get a better quantization error for small values we can switch the dynamic exponent and the base exponent. For regular dynamic quantization the base exponent is $10^0 = 1$ and each zero bit decreases the exponent by a factor of 10 for a minimum value of $10^{-7}$. We invert this starting with base $10^{-7}$ and each zero bit increases the exponent by 10 for a maximum value of 1. We denote this quantization as *inverse dynamic quantization*.

### E.2 QUANTILE QUANTIZATION: A LOSSY MINIMUM ENTROPY ENCODING

A lossy minimum entropy encoding with $k$ bits has the property that for any input data, the quantized outputs take the value of each of the $2^k$ different bit representations equally often.

More formally, a lossy minimum entropy encoding can be described in the following way. Given an infinite stream of sampled real numbers $x_i$ where $x_i$ is distributed as $X$, an arbitrary probability distribution, a lossy minimum entropy encoding is given by the $k$-bit quantization map $\mathbf{Q}^{\mathrm{map}} \in \mathbb{R}^{2^k}$ which maps values $q \in \mathbb{R}^{2^k}$ to indices $0, 1, \ldots 2^k$ which has the property that if any number of elements $x_i$ from the stream are quantized to $x_i^q$ we do not gain any information which is predictive of future $x_{j>i}^q$.

One way to fulfill this property for arbitrary probability distributions $X$, is to divide the probability distribution function $f_X$ into $2^k$ bins where each bin has equal area and the mid-points of these bins are values $q$ of the quantization map $\mathbf{Q}^{\mathrm{map}}$. Empirically, this is equivalent to a histogram with $2^k$ bins where each bin contains equal number of values.

How do we find the mid-points for each histogram bin? This is equivalent to finding the $2^k$ non-overlapping values $x$ for the cumulative distribution function $F_X$ with equal probability mass. These values can most easily be found by using its inverse function, the quantile function $Q_X = F_X^{-1}$. We can find the mid-points of each of the histogram bins by using the mid-points between $2^k + 1$ equally

spaced quantiles over the range of probabilities $[0, 1]$:

$$q_i = \frac{Q_X\left(\frac{i}{2^k+1}\right) + Q_X\left(\frac{i+1}{2^k+1}\right)}{2}, \tag{5}$$

To find $q$ empirically, we can estimate sample quantiles for a tensor $\mathbf{T}$ with unknown distribution $X$ by finding the $2^k$ equally spaced sample quantiles via $\mathbf{T}$'s empirical cumulative distribution function. We refer to this quantization as *quantile quantization*.

To estimate sample quantiles efficiently, we devise a specialized approximate quantile estimation algorithm, SRAM-Quantiles, which is more than 75x faster than other approximate quantile estimation approaches (Govindaraju et al., 2005; Dunning and Ertl, 2019). SRAM-Quantiles uses a divide-and-conquer strategy to perform sorting solely in fast SRAM. More details on this algorithm can be found in the Appendix Section F.

### E.3 Visualization: Dynamic vs Linear quantization vs quantile quantization

Figure 6 shows the mapping from each to the 255 values of the 8-bit data types to their value normalized in the range [-1, 1]. We can see that most bits in dynamic quantization are allocated for large and small values. Quantile quantization is introduced in Appendix E.2.

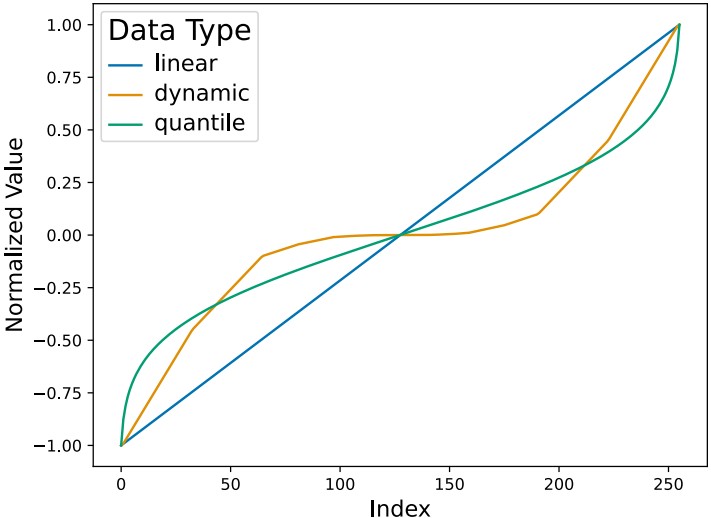

Figure 6: Visualization of the quantization maps for the linear, dynamic and quantile quantization. For quantile quantization we use values from the standard normal distribution and normalize them into the range [-1, 1].

## F   SRAM-Quantiles: A Fast Quantile Estimation Algorithm

To estimate sample quantiles of a tensor one needs to determine the empirical cumulative distribution function (eCDF) of that tensor. The easiest way to find the eCDF is to sort a given tensor. Once sorted, the quantiles can be found by using the value at index $i = q \times n$ where $i$ is the index into the sorted array, $q$ is the desired quantile and $n$ is the total elements in the tensor. While simple, this process of estimating quantiles is computationally expensive and would render training with quantile quantization too slow to be useful.

Similar to other quantile estimation approaches, our GPU algorithm, SRAM-Quantiles, uses a sliding windows over the data for fast, approximate quantile estimation with minimal resources. Greenwald and Khanna (2001)'s quantile estimation algorithm uses dynamic bin histograms over sliding windows to estimate quantiles. Extensions of this algorithm accelerate estimation by using more efficient data structures and estimation algorithms (Dunning and Ertl, 2019) or by using GPUs (Govindaraju et al., 2005). The main difference between this work an ours is that we only compute a limit

set of quantiles that are known a priori – 256, to be exact – while previous work focuses on general statistics which help to produce *any* quantile a posteriori. Thus we can devise a highly specialized algorithm which offers faster estimation.

The idea behind our algorithm comes from the fact that sorting is slow because it involves repeated loads and stores from main memory (DRAM) when executing divide-and-conquer sorting algorithms. We can significantly improve performance of quantile estimation if we restructure quantile estimation to respect memory hierarchies of the device on which the algorithm is executed.

On a GPU, programmable SRAM – known as shared memory – is 15x faster than DRAM but has a limit size of around 64 kb per core. The SRAM-Quantiles algorithm is simple. Instead of finding the full eCDF we find the eCDF for a subset of values of the tensor that fits into SRAM (about 4096 32-bit values). Once we found the quantiles for each subset, we average the quantiles atomically in DRAM.

This algorithm works, because the arithmetic mean is an unbiased estimator for the population mean and samples quantiles estimated via eCDFs are asymptotically unbiased estimators of the population quantile (Chen and Kelton, 2001). Thus the more subset quantiles we average, the better the estimate of the tensor-wide quantiles.

For estimating 256 quantiles on a large stream of numbers, our algorithm takes on average 0.064 ns to process one element in the stream, whereas the fastest general algorithms take 300 ns (Govindaraju et al., 2005) and 5 ns (Dunning and Ertl, 2019).

## G ADAGRAD COMPARISONS

While the main aim in this work is to investigate how the most commonly used optimizers, such as Adam (Kingma and Ba, 2014) and Momentum (Qian, 1999), can be used as 8-bit variants without any further hyperparameter tuning, it can be of interest to consider the behavior of our 8-bit methods under different scenarios. For example, one difference between Adam/Momentum and AdaGrad (Duchi et al., 2011) is that AdaGrad accumulates gradients statistics over the entire course of training while Adam/Momentum use a smoothed exponential decay over time. As such, this could lead to very different 8-bit quantization behavior where there are large difference between the magnitude of different optimizer states. Such large differences could induce a large quantization error and degrade performance of 8-bit optimizers.

To investigate this, we train small 209M parameter language models on the RoBERTa corpus (Liu et al., 2019). We use the AdaGrad hyperparameters introduced by Keskar et al. (2019). Results are shown in Table 7. From the results we can see that our 8-bit methods do not work as well for AdaGrad. One hypothesis is that this is due to the the wide range of gradient statistics of AdaGrad which comes from averaging the gradient over the entire course of training. To prevent poor quantization in such scenarios, stochastic rounding proved to be very effective from our initial experiments with other 8-bit optimizer. While we abandoned stochastic rounding because we did not see any benefits for Adam and Momentum, it could be an effective solution for AdaGrad. We leave such improved 8-bit quantization methods for AdaGrad to future work.

Table 7: AdaGrad compared to Adam performance for a 209M parameter language model on the RoBERTa corpus. The 8-bit methods use stable embedding layer. AdaGrad hyperparamters are taken from (Keskar et al., 2019).

| Optimizer | Valid Perplexity |
|---|---|
| 32-bit Adam | 16.7 |
| 8-bit Adam | 16.4 |
| 32-bit AdaGrad | 19.4 |
| 8-bit AdaGrad | 19.7 |

