# OpenReview forum: "8-bit Optimizers via Block-wise Quantization"
_ICLR.cc/2022/Conference — ICLR 2022 Spotlight_

### Official Review · Reviewer_zhGJ · 2021-11-02

**Correctness:** 3
**Technical Novelty And Significance:** 3
**Empirical Novelty And Significance:** 4
**Recommendation:** 6
**Confidence:** 5

**Details Of Ethics Concerns:**

There are videos, github repos and arvix preprints tagged with affiliations online following the links provided in the paper footnote.
These additional resources/information helped me understand the paper better, but I am not sure it is okay for these to help a double-blind reviewed submission.
Considering this fact, I lowered my score to 6 from 8.

**Main Review:**

This paper shows very impressive empirical results and has an open-sourced codebase which makes it reproducible. Memory saving and time-saving are especially impressive. These empirical results themselves are very valuable to the community given there are not many open-sourced codebases, to begin with.

The methodology of dynamic (tree) quantization and stable embedding are reasonable but not surprising to the quantization research community. The authors modified the dynamic tree quantization, which also appears to be reasonably motivated. Appendix E seems to support the argument despite there being few places where strong implicit assumptions were made:  mean of gradient zero over time, and constant variance. These should be thoroughly discussed in the appendix as well.

Quesion:
1.  Why image tasks do not have as much saving as NLP tasks? In table 1, Image related tasks's savings are marginal versus NLP related tasks. Is there a way to make this work for Image, speech task as well?

There are a few minor points that do not affect my score:
- Several broken reference links throughout the text with ??
- Many attempts have been made on quantizing the optimizer states from 32-bit to 8-bit using non-linear quantization, but just not as successful maybe. E.g. <Sun et al. 2019, HFP8> <Pappalardo 2021, Brevitas><Li et al. 2020 End-to-end Quantized Training via Log-Barrier Extensions>  can do such jobs as well.


**Summary Of The Paper:**

This paper proposes a non-linear block-wise quantization method to reduce the memory overhead of stateful optimizers, without sacrificing performance going from 32bits to 8 bit. The authors combine block-wise quantization with 2 methods to stabilize training: dynamic tree quantization and a stable embedding layer. Results on WMT, GLUE and Moco show the effectiveness of the method.


**Summary Of The Review:**

This paper has very valuable shared resources, and it's worthwhile for the research community to notice.

---

> ### Author Response · Authors · 2021-11-15
> **Addressing concerns/comments**
>
> **Why image tasks do not have as much saving as NLP tasks? In table 1, Image related tasks's savings are marginal versus NLP related tasks. Is there a way to make this work for Image, speech task as well?**
>
> Tensors allocated during training can have two different relationships with respect to how much memory they consume. There are tensors that only rely on the model size (parameters, parameter gradients, optimizer states) and then there are tensors that rely on model and input size (input/input gradients).
>
> In the case of convolutional networks the input/input gradients are large (large images of size 224x224) while the weights are rather small (3x3 convolution kernel). As such, the memory due to the inputs/inputs gradients is much larger than the memory due to the optimizer states. Overall, convolutional networks do not exhibit large memory savings when 8-bit optimizers are used.
> However, this is not related to the domain (e.g. vision/speech), but to the input size / model size. For example, using 8-bit Adam instead of  32-bit Adam for DALL-E would save 72 GB of memory, because it has relatively many parameters (12B) compared to its input size (256x256)
>
> Large speech models, in particular transformer models, would also exhibit a large memory saving, but 8-bit optimizers applied to a small convolutional speech model would only save a couple of hundred MB of memory.
>
> **There are a few minor points that do not affect my score: (1) Several broken reference links throughout the text with ?? [...] (2) Many attempts have been made on quantizing the optimizer states from 32-bit to 8-bit using non-linear quantization, but just not as successful maybe. E.g. <Sun et al. 2019, HFP8> <Pappalardo 2021, Brevitas><Li et al. 2020 End-to-end Quantized Training via Log-Barrier Extensions> can do such jobs as well.**
>
> We could only find a single broken reference and we fixed this reference. Thank you!
>
> We already included Sun et al 2019 in our related work, but we missed Pappalardo 2021, Brevitas><Li et al. 2020 End-to-end Quantized Training via Log-Barrier Extensions>. We will add them. They seem to perform 8-bit training with 8-bit gradients, but do not use 8-bit momentum (stateful) or 8-bit SGD (stateless).
>
> **There are videos, github repos and arvix preprints tagged with affiliations online following the links provided in the paper footnote.**
>
> We cannot find any footnote in the main paper or appendix that has any deanonymizing information. Can you please check again if there is any problem with the openreview draft and point us to the exact footnote? Thank you!

---

### Official Review · Reviewer_QSRR · 2021-11-02

**Correctness:** 3
**Technical Novelty And Significance:** 3
**Empirical Novelty And Significance:** 3
**Recommendation:** 8
**Confidence:** 5

**Main Review:**

**Pros:**
- The authors conducted a wide range of experiments on diverse NLP and computer vision tasks performing consistent memory footprint saving without performance degradation.
- The proposed method is stand-alone and potentially can be applied in parallel with other compression techniques such as weight, activation quantization, and pruning.

**Cons:**
- The proposed method is beneficial more to neural networks with a high amount of the parameters proportionally to the activations thus for convolutional neural networks the memory saving ratio is much smaller rather for transformer-based NNs.
- There are several broken references that should be fixed

**Summary Of The Paper:**

This paper addresses the very important problem of reducing the memory footprint of neural networks training. For that matter, the authors propose replacing standard optimizers with their 8-bit quantize counterparts.
The proposed scheme of optimizer state quantization has three components (I) block-wise quantization which osolat4ed outliers impact on the error (ii) dynamic quantization which quantize both small and large values with high precision and (iii) stable embedding layer which improves the stability of the optimizer during training.

**Summary Of The Review:**

The proposed paper proposed 8-bit quantized optimizers counterparts for saving memory footprints during training.
Empirical results show the proposed method's efficiency over a wide range of tasks and architectures.
I believe that the proposed method introduces an important contribution towards the efficient training of NNs.
Hence I vote for its acceptance in a pre-rebuttal phase.
***********
Post-rebuttal discussion:
I have fully satisfied with the author's feedback and revision version of the paper according to my and other reviewers' concerns.
My score remains "8" and I am convinced that this manuscript would significantly contribute to ML community, hence I vote for his acceptance.

---

> ### Author Response · Authors · 2021-11-15
> **Addressing concerns/comments**
>
> **The proposed method is beneficial more to neural networks with a high amount of the parameters proportionally to the activations thus for convolutional neural networks the memory saving ratio is much smaller rather for transformer-based NNs.**
>
> This is true. We included the convolutional neural network results to show that 8-bit optimizers do not degrade performance for convolutional architectures. While the benefits for convolutional layers are minimal, our results gives us confidence that 8-bit optimizers work for mixed architectures, such as Primer [1], which is a regular transformer that has a self-attention block that consists of both convolutional and regular transformer layers.
>
> **There are several broken references that should be fixed**
>
> We inspected our draft and could only find a single broken reference. We fixed that reference – thank you!

---

### Official Review · Reviewer_6PcJ · 2021-11-02

**Correctness:** 3
**Technical Novelty And Significance:** 3
**Empirical Novelty And Significance:** 3
**Recommendation:** 8
**Confidence:** 5

**Main Review:**

```Updated Score```: See my comments why: https://openreview.net/forum?id=shpkpVXzo3h&noteId=PWzRo82xR07 and hopefully authors address the additional comments in the final version.

Paper proposes block-wise quantization (dynamic) to reduce the states (momentum and second moment) in diagonal first order optimizers and successfully implements a 8-bit Adam implementation that works as well as its f32 variant while being memory efficient. Paper further identifies embedding layers as a source of instability and proposes layer norm to improve stability,  and leaves optimizer states in f32 for this layer.

Strengths:
+ Efficient 8bit implementation of Adam on GPUs
+ Careful work to reduce quantization errors, and improvement to existing algorithms (for example quantile estimation)
+ Blockwise quantization is very neat way to deal with outliers

Weakness/Improvements:
+ Lack of comparison to previously established SOTA for low memory optimization (See comments on AdaFactor)
+ Results are at similarish (lower) batch sizes. It would have been interesting to see the method is applicable for large batch training. One could conjecture that heavy tailed nature noise (https://arxiv.org/abs/1912.03194) and its associated effects on the diagonal preconditioner might make quantization harder.
+ Ignores AdaGrad line of work

==AdaFactor comparisons==

AdaFactor has the option to reduce the memory used by momentum states completely by replacing it with the cheaper adaptive gradient clipping, and has been used for large model training (see https://arxiv.org/pdf/2006.16668.pdf for hyper-parameters. (See beta1=0))
[1] https://github.com/tensorflow/lingvo/blob/master/lingvo/core/optimizer.py#L1044

The paper includes the f32 variant, but do not include the bfloat16 variant [1] comparison
[2] https://github.com/tensorflow/tensor2tensor/blob/master/tensor2tensor/utils/adafactor.py See parameter encoding.

== AdaGrad and its variants ==
Paper currently misses citation of the entire AdaGrad-line of work, and comparision/implementation, I hope the authors can address these easily.

[1] https://www.jmlr.org/papers/volume12/duchi11a/duchi11a.pdf
[2] Memory efficient adaptive optimization https://proceedings.neurips.cc/paper/2019/file/8f1fa0193ca2b5d2fa0695827d8270e9-Paper.pdf
[3] https://openreview.net/pdf?id=SklKcRNYDH (ICLR 2020)

Does the proposed block quantization work well for AdaGrad variants as well? Reason to ask is, after reading the paper, it's not entirely clear if the proposed approach works for AdaGrad, SM3 and Extreme Tensoring as it accumulates statistics over the entire horizon and has a very different range of values (distribution of values) than Adam's second-moment statistics.

It would be valuable to the community if authors could address this in the paper by demonstrating improvements or showing negative results and challenges to this end.  For example, in this work authors leave compressing states for embedding layers (with layernorm) as future work.

== Some related citations for block wise quantization ==
For weights of a network; similar idea of block-wise quantization (but for binarization) has been employed in "Improving Bi-Real Net with block-wise quantization and multiple-steps binarization on activation", Duy H. Le; Tuan V. Pham



**Summary Of The Paper:**

The paper shows a working implementation of 8bit states for momentum and second momentum. This is achieved by using block-wise dynamic quantization for efficient compression and fast implementation. They show drastic improvement over f32 diagonal optimizers (mainly Adam) and improvement over a subset of previously proposed sub-linear memory optimizer (AdaFactor f32 variant).

**Summary Of The Review:**

Authors propose 8-bit optimizers for training neural networks, and demonstrated its usefulness in one regime. There are several important ablations (mentioned in the reviews, and I will revise the score accordingly) that are missing which are important it ascertain its wide utility, as well understand its limitations.

---

> ### Author Response · Authors · 2021-11-15
> **Addressing Adafactor, AdaGrad (new results), and batch size comments**
>
> TL;DR (concerning Adafactor memory efficiency and AdaGrad). We will add significant discussion and have run some new comparisons, thanks for the pointers. Our main claims and results remain unchanged.  In particular we want to emphasize (longer version with full details below):
> - The main purpose of our comparison with Adafactor is to establish performance for a factorization approach and compare with 8-bit quantization. Our method can be combined seamlessly with Adafactor (or any other optimizer) to make more memory efficient optimizers.
> - We report initial 8-bit vs 32-bit AdaGrad results and will discuss AdaGrad in the related work and appendix. We did not design our 8-bit optimizer procedure to be compatible with AdaGrad out-of-the box, but we think this is possible through stochastic rounding.
>
> **Lack of comparison to previously established SOTA for low memory optimization (See comments on AdaFactor)**
>
> Thank you for pointing out work that uses Adafactor without the first moment and with a factorized second moment. We will include a discussion of this work in our related work section. It is difficult to compare against this result because it was trained on private data. We are not aware of other work that uses this optimizer configuration.
>
> Our comparison with Adafactor was mostly meant to compare the drop in performance between optimizer state factorization and 8-bit quantization rather than state-of-the-art memory compression. Using 16-bit Adafactor in our experiments would decrease the memory footprint, but would not improve the drop in performance due to decreased precision. While 16-bit Adafactor would have a similar memory footprint to 8-bit Adam, 8-bit Adafactor would also be possible. We will update our work and highlight that these versions of Adafactor will lead to comparable or better memory footprint compared to 8-bit Adam.
>
> **Results are at similarish (lower) batch sizes. It would have been interesting to see the method is applicable for large batch training. [..]**
>
> For our large scale language modeling experiments with 1.5B parameters we use a batch size of 2 million tokens. We would consider this fairly large and the results show that 8-bit Adam performs just as well as 32-bit Adam. With our current resources it would be difficult to run experiments that run long enough to yield meaningful results with batch sizes larger than 2 million tokens.
>
> **Ignores AdaGrad line of work**
>
> Thank you for pointing out the AdaGrad line of work. We will add a discussion. We mainly compared against compressed optimizers for which many large-scale results are known in the literature (Adam and Adafactor). Most AdaGrad implementations use the dense (non-diagonal) implementation [1,2] and do not provide a memory benefit compared to similar optimizers such as RMSProp. While other optimizers like SM3 and methods such as extreme tensoring provide memory benefits these methods have not been used in large scale training outside of the main paper.
>
> Nevertheless, we are happy to include a discussion of AdaGrad and related methods to make readers aware that theoretically such methods yield a memory footprint similar or smaller than 8-bit Adam.
> [1] https://github.com/pytorch/pytorch/blob/master/torch/optim/adagrad.py#L68
> [2]https://github.com/keras-team/keras/blob/master/keras/optimizer_experimental/adagrad.py#L82-L86
>
> **Does the proposed block quantization work well for AdaGrad variants as well?**
>
> Thank you for this interesting question. We did not consider this because we did not intend for 8-bit optimizers to be used with AdaGrad. Nevertheless, we have the following observations from related experiments with stochastic rounding for cases where the range of optimizer statistics is very wide – the same scenario AdaGrad. In these cases, stochastic rounding recovers full 32-bit performance if lots of very small values are routinely quantized to zero.
>
> We also ran some baselines to get some hard results for 8-bit AdaGrad. We ran three random seeds with the Adagrad setup from Keskar et al., 2019 [1] on the RoBERTa corpus: The perplexities for each random seed are as follow:
>
> - 32-bit AdaGrad: 19.44, 19.46, 19.03
> - 8-bit AdaGrad: 19.75, 19.72, 19.59
>
> As such, 8-bit AdaGrad performs slightly worse than 32-bit AdaGrad. In general, these perplexities are rather poor, which might indicate that AdaGrad is not competitive with AdaFactor/Adam or that our hyperparameters were poorly chosen. It is out of scope of this rebuttal to do an extensive hyperparameter search.
>
> So 8-bit AdaGrad would require adaptations but we think 32-bit performance could be recovered with stochastic rounding. We will include this discussion in the appendix.
>
> [1] CTRL: A Conditional Transformer Language Model for Controllable Generation
>
> **== Some related citations for block wise quantization ==**
> Thank you, we will add it to our related work.

---

> > ### Comment · Reviewer_6PcJ · 2021-11-15
> > **Thanks for the detailed response.**
> >
> > Here are some quick replies to the comments from authors, as I thought it might help the authors with the revisions. Thanks to the authors for the detailed response.
> >
> > 1. Large batch size and computational resources: I agree that it would be more resource-intensive for 1.5B parameters; optionally, Authors can try 8-bit Adam for BERT-Base (or Large)/encoder models with 64k batch size (see SM3 paper example for experiments), and would strengthen the paper. However, this is optional.
> >
> > 2. AdaGrad/SM3/Extreme Tensoring: These techniques indeed work and requires adding momentum (EMA/HB See SM3 (https://github.com/google-research/google-research/blob/master/sm3/sm3.py#L159) or arxiv.org/abs/2002.11803) on-top of the preconditioned gradients to work well in terms of test PPL. It would be advantageous to know the results of experiments with momentum for AdaGrad. The additional discussions you mentioned sound good.
> >
> > 3. Work that uses beta=0.0 (using adaptive gradient clipping) for AdaFactor beyond GShard are  Multilingual nmt https://arxiv.org/pdf/1907.05019.pdf, T5 https://arxiv.org/abs/1910.10683, ChatBot https://arxiv.org/pdf/2001.09977.pdf
> >
> > 4. New feedback based on recent observations: One comment about stable embedding layer is that the practice of using layer norm on top of embeddings dates back to BERT (from Devlin et al) -- this is something the paper should discuss/cite.

---

> > > ### Author Response · Authors · 2021-11-17
> > > **Thank you for additional questions and comments!**
> > >
> > > **Large batch size and computational resources: I agree that it would be more resource-intensive for 1.5B parameters; optionally, Authors can try 8-bit Adam for BERT-Base (or Large)/encoder models with 64k batch size (see SM3 paper example for experiments), and would strengthen the paper. However, this is optional.**
> > >
> > > Thank you for suggesting further experiments. The scaling laws paper [1] says that the best batch size for their largest models (175B parameters) is between 1-2M tokens per mini-batch -- exactly the batch size used in our large-scale experiment. A batch size of 64k would be equivalent to 1024*64k = 65M tokens per mini-batch. While such an experiment would provide theoretical insights into how our methods behave for such large batch sizes, our work is meant to show-case the behavior of 8-bit optimizers for more commonly used training setups. If other reviewers agree this is an important experiment, we are happy to run a basic version of this.
> > >
> > > [1] Scaling Laws for Neural Language Models
> > >
> > > **AdaGrad/SM3/Extreme Tensoring: These techniques indeed work and requires adding momentum (EMA/HB See SM3 (https://github.com/google-research/google-research/blob/master/sm3/sm3.py#L159) or arxiv.org/abs/2002.11803) on-top of the preconditioned gradients to work well in terms of test PPL. It would be advantageous to know the results of experiments with momentum for AdaGrad. The additional discussions you mentioned sound good.**
> > >
> > > Thank you for suggesting possible follow-up experiments based on our new 8-bit AdaGrad results. There exist more than 15 commonly used optimizers [1], and certainly, for some optimizers our 8-bit method will fail. We do not claim to have exhaustively shown that our 8-bit method works for all optimizers or that you couldn’t design an extension of AdaGrad that would be better. Our 8-bit methods were designed  to work well with Adam and Momentum, which we hope we have shown convincingly. Applying it to other optimizers is an interesting area for future work that we hope others will be inspired to do.
> > >
> > > [1] Descending through a Crowded Valley - Benchmarking Deep Learning Optimizers
> > >
> > > **Work that uses beta=0.0 (using adaptive gradient clipping) for AdaFactor beyond GShard are Multilingual nmt https://arxiv.org/pdf/1907.05019.pdf, T5 https://arxiv.org/abs/1910.10683, ChatBot https://arxiv.org/pdf/2001.09977.pdf**
> > >
> > > Thank you for pointing out other papers that might use AdaFactor with beta=0.0. However, reading these papers, we cannot find a reference to this particular setup (one paper mentions “factorized momentum”, but this is undefined). We are unable to discuss this, because from the publicly available information it is unclear how AdaFactor is used in these works.
> > >
> > > **New feedback based on recent observations: One comment about stable embedding layer is that the practice of using layer norm on top of embeddings dates back to BERT (from Devlin et al) -- this is something the paper should discuss/cite.**
> > >
> > > Please have a look at appendix B where we discuss this. To summarize the appendix section: Yes, many frameworks and models use a layer norm directly after the embedding layer, but this is not standard. Models such as RoBERTa, BART, RAG, and many other commonly used models do not use a layer norm after the embedding layer. As such, we find it critical to mention this detail which will be new for many researchers. We will extend the discussion to make it clearer that for some frameworks this is already standard.

---

> > > > ### Comment · Reviewer_6PcJ · 2021-11-17
> > > > **Feedback on the responses**
> > > >
> > > > == Scaling laws ==
> > > > If authors read my comment again; it was to directly compare this technique for large scale training of BERT (100-300M) parameter network, not 175B parameter network. See https://arxiv.org/abs/2102.06356 https://arxiv.org/abs/1904.00962, and Moreover, the hypothesis was to test limitation/applicability of this technique in regime where preconditioner distribution changes.
> > > >
> > > > == Standard configs for AdaFactor ==
> > > > "Cannot find a reference"
> > > > A quick search for script/configs for AdaFactor to reproduce T5 in their official codebase found this:
> > > > https://github.com/google-research/text-to-text-transfer-transformer/tree/main/t5/models/gin/models
> > > > Bi-transformer instantiates AdaFactor with default params, including setting beta1=0.0 and saving all the states for momentum.
> > > >
> > > > Critically, it's mentioned in the original paper:  https://arxiv.org/abs/1804.04235, and they successfully replace momentum with adaptive clipping.
> > > >
> > > > == Crowded valley ==
> > > > I do not understand the arguments here. Yes, there are 100's of optimizers that are variants of diagonal preconditioners and primarily based on 3 key optimizers, AdaGrad -> RmsProp -> Adam.
> > > >
> > > > My question was why AdaGrad line of algorithms  (previously published work) which have been shown to reduce memory were ignored in the paper; Primary motivation is to know if the approach has any limitations when computing preconditioners over the entire horizon. Note, the question is around understanding the limitations, if any. The question is not about arbitrary variants of Adam/AdaGrad but very specific about the proposed technique empirically works only for moving averaged statistics vs. entire horizon. As authors ran AdaGrad comparison already, my motivation for asking authors to add momentum was to improve their own results!
> > > >
> > > > == LayerNorm and stability ==
> > > > Thanks for the discussion, but my point stands. It would be good to cite/discuss that this practice has been empirically deployed in the latest architectures (even though it's not called out).
> > > >
> > > > == New feedback on Appendix B ==
> > > > The section mentions instability is due to large gradients due to the sparsity of inputs, and the gradients can be 100x larger for some rare tokens.  However,  techniques such as Adam, where the preconditioned gradient is formed by Momentum(g) / sqrt(EMA(g^2)) + epsilon, is scale-invariant (modulo epsilon). Thus, it leads to more questions on ablations:
> > > > (1) can this by fixed by simply scaling epsilon for tokens based on frequency (or raising the fixed epsilon to be higher)
> > > > (2) using AdaGrad (designed to work better under sparsity) with momentum where it is guaranteed that the learning rate continues to decrease.

---

> > > > > ### Author Response · Authors · 2021-11-19
> > > > > **Thank you for the discussion!**
> > > > >
> > > > > Thanks for the interest and discussion! We have chosen to focus on developing 8-bit variants of Adam and Momentum, as they are by far the most popular methods for training transformers. We have shown strong results over a very varied set of language and vision tasks, and believe these will be of interest to the community.
> > > > >
> > > > > While we have not yet been able to achieve a competitive memory/performance trade-off with Adafactor or Adagrad, we will be happy to follow your suggestions for hyperparameters, and will update the paper if we find improved results. We will of course discuss these optimizers in more detail. We agree that developing 8-bit versions of these and other optimizers would be interesting, but we leave this to future work.

---

### Author Response · Authors · 2021-11-23
**Rebuttal Revision Summary**

We thank all reviewers for their valuable feedback! The new revision of our draft addresses the concerns of the reviewers. Changes are marked in blue. We made the following changes:

Main paper:
- added a discussion of AdaGrad in the related work section
- added missing references to SM3, extreme tensoring, Brevitas, and End-to-end Quantized Training via Log-Barrier Extensions
- added a missing reference to the stable embedding layer section
- extended discussion on Adafactor with beta1=0.0
- removed the broader impacts section to respect the page limit

Appendix:
- added initial AdaGrad experiments and their discussion
- added extended discussion on related work of the stable embedding layer and that its components

---

> ### Comment · Reviewer_6PcJ · 2021-11-23
> **Thanks for adding additional discussion**
>
> As a reviewer, I am happy that authors have added relevant discussions to related work, and given credit to empirical practice (Adafactor beta1=0.0 and Stable embedding layer in BERT from Devlin et al).
>
> I have few comments on the updated paper that I hope Authors address/acknowledge.
>
> 1. AdaFactor beta1=0.0, citation should go to the Shazeer & Stern 2018 not Lepikhin 2020, if the authors intend to show it as common practice, please cite all the relevant papers as some of them predate Lepikhin 2020.
>
> 2. Please add discussion to "ADAGRAD COMPARISONS" about including 8-bit momentum as avenue to improve AdaGrad perplexity. See arxiv.org/abs/2002.11803 (Adagrad performs better on Transformers once momentum is included).
>
> There are still some relevant open discussions from our thread:
>
> 3. The weakest part of the paper is the `Stable embedding layer` discussion, as authors conjecture it is due to sparsity/power law of the feature frequency, but does not ablate or dig into the specifics, and concerning part is that Adam indeed normalizes the update to have a constant learning rate (so there is no 100x larger update), and perhaps a better solution would have been AdaGrad for embedding layers or changing epsilon to be larger while multiplying the gradient by a constant which is an increasing warmup schedule to make epsilon smaller over time (both of which are empirical practice in models that I as a reviewer work on).
>
> 4. Also a note for future work, it would be interesting to see under hyper-parameter tuning which works better,  0-bit momentum (beta1=0.0, adaptive gradient clipping) + adafactor or 8-bit Adam or 8-bit AdaGrad(with 8-bit momentum) or 8-bit momentum, + sm3.
>
> Finally, I will update my score to an Accept (8) based on the fact that this paper proposes a useful/practical contribution (8-bit Adam) for accelerating the recent endeavor that community has been engaging on, which is the training of largest models possible under hardware constraint of the day.

---

> > ### Author Response · Authors · 2021-11-29
> > **Thank you!**
> >
> > Thank you for the helpful discussion! We will correct and extend (1) and (2) in our next draft. We agree that the optimizers suggested in (4) would be very insightful and interesting for future work. Since we open-source our software, we hope it will be easy for the community to add these optimizers.
> >
> > Thank you for extending the discussion about (3) the stable embedding layer. There are many factors and confounding variables to consider. It also changes dramatically at scale, making it particularly difficult to control for. We will add language modeling ablations for all stable embedding layer components to the appendix of the final draft. We like your idea about AdaGrad for just the embedding layer — it indeed fits our assumptions very well! We will run this experiment and will add it to the AdaGrad or stable embedding section in the appendix.

---

### Decision · Program_Chairs · 2022-01-20

**Decision:**

Accept (Spotlight)

**Comment:**

This paper proposes Adam and Momentum optimizers, where the optimizer state variables are quantized to 8bit using block dynamics quantization. These modifications significantly improve the memory requirements of training models with many parameters (mainly, NLP models). These are useful contributions which will enable training even larger models than possible today. All reviewers were positive.